# Electrochemical Radical Tandem Difluoroethylation/Cyclization of Unsaturated Amides to Access MeCF_2_-Featured Indolo/Benzoimidazo [2,1-*a*]Isoquinolin-6(5*H*)-ones

**DOI:** 10.3390/molecules29050973

**Published:** 2024-02-22

**Authors:** Yunfei Tian, Dongyu Guo, Luping Zheng, Shaolu Yang, Ningning Zhang, Weijun Fu, Zejiang Li

**Affiliations:** 1Key Laboratory of Fuction-Oriented Porous Materials of Henan Province, College of Chemistry and Chemical Engineering, Luoyang Normal University, Luoyang 471934, Chinawjfu@lynu.edu.cn (W.F.); 2Key Laboratory of Medicinal Chemistry and Molecular Diagnosis of the Ministry of Education, State Key Laboratory of New Pharmaceutical Preparations and Excipients, Key Laboratory of Chemical Biology of Hebei Province, College of Chemistry & Materials Science, Hebei University, Baoding 071002, Chinalizejiang898@126.com (Z.L.)

**Keywords:** electrochemical synthesis, difluoroethylation, radical, polycyclic compounds, tandem cyclization

## Abstract

A metal-free electrochemical oxidative difluoroethylation of 2-arylbenzimidazoles was accomplished, which provided an efficient strategy for the synthesis of MeCF_2_-containing benzo[4,5]imidazo[2,1-*a*]-isoquinolin-6(5*H*)-ones. In addition, the method also enabled the efficient construction of various difluoroethylated indolo[2,1-*a*]isoquinolin-6(5*H*)-ones. Notably, this electrochemical synthesis protocol proceeded well under mild conditions without metal catalysts or exogenous additives/oxidants added.

## 1. Introduction

Organofluorine compounds are very useful and attractive organic molecules, which play pivotal roles in pharmaceuticals, agrochemicals, and performance materials [1,2,3,4,5,6]. Among various fluorine-containing groups, the difluoroethyl group (CF_2_Me) is receiving more and more attention due to its special chemical and biological properties [7,8,9]. In particular, the introduction of a difluoroethyl group can improve the metabolic stability and potency of a target molecule [10,11]. Therefore, the development of new and efficient protocols for the rapid introduction of the difluoroethyl group to target compounds is in high demand. In this context, the radical strategy has emerged as a powerful approach for the synthesis of CF_2_Me-containing compounds. However, recent developments in this direction have always involved peroxides or photoredox catalysts [12,13,14,15,16]. On the other hand, electrochemical synthesis has also attracted much attention due to the advantages of avoiding the usage of chemical oxidants and reductants [17,18,19,20,21,22,23]. Very recently, much progress in electrochemical difluoroethylation has been made by Hu [24] and our group [25,26]. Although these methods have provided innovative transformations, the preparation of CF_2_Me-substituted polycyclic compounds has not been achieved.

Benzo[4,5]imidazo[2,1-*a*]isoquinolines and indolo [2,1-*a*]isoquinolines, which are significant classes of fused polycyclic nitrogen-containing scaffolds, widely exist in natural products and pharmaceuticals (Figure 1(1)) [27,28,29,30,31]. Although substantial efforts have been contributed to the construction of these polycyclic compounds [32,33,34,35,36,37,38,39], the CF_2_Me-containing target polycycles remain a great challenge to date (Figure 1(2)). As part of our continuing interest in difluoroethylation functionalizations, we anticipated that the radical cyclization process would provide a feasible platform for the synthesis of CF_2_Me-revised target polycycles. Herein, we report an electrochemical-induced radical cascade cyclization strategy, whereby a series of CF_2_Me-substituted indolo [2,1-*a*]isoquinolines and benzo[4,5]imidazo[2,1-*a*]isoquinolines could be efficiently prepared under mild and chemical oxidant-free conditions (Figure 1(3)).

## 2. Results and Discussion

We began our investigation by examining the electrochemical difluoroethylation reaction of *N*-methacryloyl-2-phenylbenzoimidazole (**1a**) with NaSO_2_CF_2_Me (**2a**) (Table 1). To our delight, the electrolysis furnished an 86% yield of the desired cyclization product **3a** at a constant voltage of 2.1 V in an undivided cell equipped with a carbon plate anode and a Pt plate cathode (Table 1, entry 1). Then, the various conditions, such as voltage, electrode material, electrolyte, and solvent, were measured. Neither lower voltage nor higher voltage led to a higher efficiency (entries 2,3). C(+)|Pt(−) was proved to be the optimal electrode material combination compared to others (entries 4–6). A switch of the electrolytes LiClO_4_ to other electrolytes such as Et_4_NClO_4_, *^n^*Bu_4_NClO_4_, or *^n^*Bu_4_NPF_6_ significantly restrained the reaction (entries 7–9). The change in solvent proportion failed to improve the yield of **3a** (entries 10, 11). The cyclization product **3a** could not be observed without electricity (entry 12).

With the above-optimized conditions in hand, we investigated the substrate scopes (Figure 1). It can be seen that a wide range of 2-arylbenzoimidazoles with either electron-donating or electron-withdrawing substituents worked well and afforded the corresponding products in good to high yields (**3a**–**3h**). The ortho-substituted 2-arylbenzoimidazoles were also tolerated in the reaction, and the desired products were obtained at 70–88% yields (**3i**–**3m**), among which the single crystal structure of product **3m** was obtained. When utilizing meta-substituted 2-arylbenzoimidazoles as the starting materials, the reactions demonstrated good site-selectivity with no regioisomers detected (**3n**). It also occurred smoothly on disubstituted substrates to produce the cyclization products in good yields (**3o** and **3p**). The 3, 5-di substituent was successfully converted to the target product **3q** at 72% yield without the interference of steric hindrance. For Ar^1^ substituents, the dimethyl-substituted *N*-methacryloyl-2-phenylbenzoimidazoles gave the desired products **3r** in good yields. The substrates with phenyl- or benzyl-substitution of the terminal olefin were also able to produce the relevant products (**3s** and **3t**). The substrates containing naphthalene or thiophene were all compatible with this reaction mode, delivering the corresponding products **3u** and **3v** at 74% and 67% yields, respectively.

Subsequently, we turned our attention to the synthesis of indolo[2,1-*a*]isoquinoline derivatives, which are key structural skeletons of various pharmaceuticals. As shown in Figure 2, the desired CF_2_Me-substituted indolo[2,1-*a*]isoquinoline derivatives were obtained in moderate to excellent yields. The halosubstituted (F^−^, Cl^−^, Br^−^) substrates also smoothly underwent a cyclization process to provide the corresponding products with good efficiency (**5b**–**5e**). Moreover, the substrates with an ethyl group at the C3 position of the indole ring were demonstrated to be suitable substrates to provide the final products at 65–75% yield (**5f**–**5h**). Notably, this method also enabled the access to cyclopropyldifluoromethylated indolo[2,1-*a*]isoquinoline (**5i**), which was further confirmed by X-ray crystallography.

Then, some control experiments were carried out to investigate the mechanism of this reaction (Figure 3). When the radical scavenger 2,6-di-tert-butyl-4-methylphenol (BHT) was added to the standard reaction system, the reaction was significantly suppressed, and the desired product was not detected by TLC. When adding 1,1-diphenylethylene into the system, only 16% yield of the target product was obtained and the radical adduct was found by HRMS. The two experiments suggested that the reaction may involve a radical pathway. To further understand the details of this reaction mechanism, cyclic voltammetry (CV) experiments were performed. As shown in Figure 2, the oxidation peaks of **1a** and **2a** were at 1.47 V and 0.67 V, respectively. These results indicated that **2a** was more easily oxidized than **1a**.

Based on the above results, a plausible mechanism of this reaction was proposed (Figure 4). First, the anodic oxidation of MeCF_2_SO_2_Na generated the MeCF_2_SO_2_ radical, which then liberated SO_2_ to afford the MeCF_2_ radical. Subsequently, the addition of the MeCF_2_ radical to the double bond of **1a** yielded a carbon-centered radical **A**. The intermediate **A** underwent further intramolecular radical cyclization to afford the aryl radical **B**. The intermediate **B** was oxidized at the anode to give aryl cation **C**, which resulted in the expected product **3a** via a deprotonation process.

## 3. Materials and Methods

### 3.1. General Methods

^1^H and ^13^C NMR and ^19^F NMR spectra were recorded on a Bruker advance Ⅲ 400 or 500 spectrometer (Billerica, MA, USA) in CDCl_3_ with TMS as the internal standard. High-resolution mass spectral analysis (HRMS (TOF)) data were measured on a Bruker Apex II. All products were identified by ^1^H, ^19^F, ^13^C NMR, and HRMS. The starting materials were purchased from Energy, J&K Chemicals (Beijing, China) or Aldrich (St. Louis, MO, USA) and used without further purification. The conversion was monitored by thin-layer chromatography (TLC). Flash column chromatography was performed over silica gel (200–300 mesh). Cyclic voltammetry experiments were carried out in an electrochemical workstation (CHI660E, Shanghai, China). 2-arylbenzimidazoles/2-arylindoles were prepared according to the reported procedures [35].

### 3.2. General Procedure for the Reaction

A 20 mL test tube with a stir bar was charged with, 2-arylbenzimidazoles/2-arylindoles (1 equiv., 0.2 mmol), MeCF_2_SO_2_Na, or sodium cyclopropyldifluoromethylsulfinate (3 equiv., 0.6 mmol), LiClO_4_ (0.3 M), MeCN (4.5 mL), H_2_O (1.5 mL). The tube was equipped with a carbon plate (10 mm × 10 mm × 3 mm) as the anode and a platinum plate (10 mm × 10 mm × 0.2 mm) as the cathode. The reaction mixture was electrolyzed in an undivided cell at room temperature under a constant voltage of 2.1 V for 3h. Upon completion, the mixture was extracted with EtOAc (10 mL × 3). The combined organic phases were dried over Na_2_SO_4_ and condensed under vacuum. The residue was purified by silica gel column chromatography to afford the final products (^1^H NMR, ^19^F NMR, and ^13^C NMR of compounds (**3a**–**v and 5a**–**i**) are shown in Appendix A).

*5-(2,2-Difluoropropyl)-5-methylbenzo [4,5]imidazo [2,1-a]isoquinolin-6(5H)-one* (**3a**) [39]. A white solid after purification by flash column chromatography (petroleum ether/ethyl acetate = 10/1), mp 153–154 °C, 56.1 mg, 86 % yield. ^1^H NMR (400 MHz, CDCl_3_): *δ* 8.52–8.49 (m, 1H), 8.37–8.35 (m, 1H), 7.84–7.82 (m, 1H), 7.59–7.55 (m, 1H), 7.50 (t, *J* = 7.2 Hz, 2H), 7.47–7.40 (m, 2H), 3.30–3.18 (m, 1H), 2.79–2.67 (m, 1H), 1.72 (s, 3H), 1.34 (t, *J* = 18.8 Hz, 3H). ^13^C{^1^H} NMR (100 MHz, CDCl_3_): *δ* 172.2, 149.6, 144.0, 140.0, 131.5, 131.3, 127.9, 126.8, 126.0, 125.9, 125.6, 122.6 (t, *J* = 240.5 Hz), 122.3, 119.8, 115.7, 48.2 (t, *J* = 23.9 Hz), 45.5, 31.1, 24.7 (t, *J* = 27.3 Hz). ^19^F NMR (471 MHz, CDCl_3_): *δ* −86.07–−86.30 (m, 2F). HRMS (ESI-TOF) *m*/*z*: Calcd for C_19_H_16_F_2_N_2_O (M + H)^+^ 327.1303; found 327.1306. IR (KBr) ν 3402, 3057, 2931, 1943, 1840, 1700, 1619, 1547, 985, 803, 644, 549 cm^−1^.

*5-(2,2-Difluoropropyl)-3,5-dimethylbenzo [4,5]imidazo [2,1-a]isoquinolin-6(5H)-one* (**3b**). A white solid after purification by flash column chromatography (petroleum ether/ethyl acetate = 10/1), mp 132–133 °C, 55.7 mg, 82 % yield. ^1^H NMR (500 MHz, CDCl_3_): *δ* 8.37 (d, *J* = 8.0 Hz, 1H), 8.35–8.33 (m, 1H), 7.80–7.79 (m, 1H), 7.44–7.38 (m, 2H), 7.30 (d, *J* = 8.5 Hz, 1H), 7.26 (s, 1H), 3.21 (q, *J* = 15.0 Hz, 1H), 2.70 (q, *J* = 15.5 Hz, 1H), 2.45 (s, 3H), 1.70 (s, 3H), 1.32 (t, *J* = 19.0 Hz, 3H). ^13^C{^1^H} NMR (125 MHz, CDCl_3_): *δ* 172.3, 149.8, 144.1, 141.8, 140.1, 131.4, 129.0, 127.1, 126.0, 125.8, 125.3, 122.5 (t, *J* = 240.6 Hz), 119.7, 119.6, 115.6, 48.2 (t, *J* = 24.1Hz), 45.5, 31.0, 24.7(t, *J* = 27.4 Hz), 21.8. ^19^F NMR (471 MHz, CDCl_3_): *δ* −86.01–−86.20 (m, 2F). HRMS (ESI-TOF) *m*/*z*: Calcd for C_20_H_18_F_2_N_2_O (M + H)^+^ 341.1460; found 341.1461. IR (KBr) ν 3415, 3068, 2964, 2361, 1909, 1715, 1613, 1557, 1238, 915, 801, 615 cm^−1^.

*5-(2,2-Difluoropropyl)-3-fluoro-5-methylbenzo [4,5]imidazo [2,1-a]isoquinolin-6(5H)-one* (**3c**). A white solid after purification by flash column chromatography (petroleum ether/ethyl acetate =10/1), mp 134–135 °C, 58.5 mg, 85 % yield. ^1^H NMR (400 MHz, CDCl_3_): *δ* 8.50 (dd, *J* = 8.4, 6.0 Hz, 1H), 8.34 (d, *J* = 7.6 Hz, 1H), 7.80 (d, *J* = 7.6 Hz, 1H), 7.46–7.39 (m, 2H), 7.22–7.15 (m, 2H), 3.29–3.17 (m, 1H), 2.71–2.59 (m, 1H), 1.70 (s, 3H), 1.40 (t, *J* = 18.8 Hz, 3H). ^13^C{^1^H} NMR (100 MHz, CDCl_3_): *δ* 171.6, 164.6 (d, *J* = 252.5 Hz), 148.8, 144.0, 142.9 (d, *J* = 7.9 Hz), 131.4, 128.5 (d, *J* = 9.1Hz), 125.7 (d, *J* = 37.5 Hz), 122.4 (t, *J* = 240.6 Hz), 119.7, 118.89, 118.87, 115.9 (d, *J* = 22.4 Hz), 115.6, 113.7 (d, *J* = 23.2 Hz), 48.3 (t, *J* = 23.7 Hz), 45.7, 30.9, 24.7 (t, *J* = 27.3Hz). ^19^F NMR (471 MHz, CDCl_3_): *δ* −85.51–−86.22 (m, 1F), −87.32–−88.02 (m, 1F), −106.94–-106.99 (m, 1F). HRMS (ESI-TOF) *m*/*z*: Calcd for C_19_H_15_F_3_N_2_O (M + H)^+^ 345.1209; found 345.1211. IR (KBr) ν 3380, 3111, 3068, 2988, 2920, 2877, 2620, 2370, 1956, 1913, 1802, 1715, 1625, 1566, 1008, 986, 786, 655, 578 cm^−1^.

*3-Chloro-5-(2,2-difluoropropyl)-5-methylbenzo [4,5]imidazo [2,1-a]isoquinolin-6(5H)-one* (**3d**). A white solid after purification by flash column chromatography (petroleum ether/ethyl acetate = 10/1), mp 174–175 °C, 62.6 mg, 87 % yield. ^1^H NMR (400 MHz, CDCl_3_): *δ* 8.43 (d, *J* = 8.4 Hz, 1H), 8.34 (d, *J* = 6.8 Hz, 1H), 7.81 (d, *J* = 8.0 Hz, 1H), 7.47–7.42 (m, 4H), 3.28–3.16 (m, 1H), 2.73–2.61 (m, 1H), 1.71 (s, 3H), 1.40 (t, *J* = 18.8 Hz, 3H). ^13^C{^1^H} NMR (100 MHz, CDCl_3_): *δ* 171.5, 148.7, 143.9, 141.8, 137.5, 131.4, 128.5, 127.4, 127.0, 126.0, 125.8, 122.5 (t, *J* = 239.0 Hz), 120.9, 119.8, 115.6, 48.2 (t, *J* = 23.6 Hz), 45.5 (d, *J* = 3.3Hz), 30.9, 24.8 (t, *J* = 27.2 Hz). ^19^F NMR (471 MHz, CDCl_3_): *δ* −85.56–−86.27 (m, 1F), −87.33–−88.03 (m, 1F). HRMS (ESI-TOF) *m*/*z*: Calcd for C_19_H_15_ClF_2_N_2_O (M + H)^+^ 361.0914; found 361.0915. IR (KBr) ν 3355, 3005, 2923, 2851, 2369, 1916, 1720, 1577, 1450, 1356, 1076, 902, 835, 765, 619 cm^−1^.

*3-Bromo-5-(2,2-difluoropropyl)-5-methylbenzo [4,5]imidazo [2,1-a]isoquinolin-6(5H)-one* (**3e**). A white solid after purification by flash column chromatography (petroleum ether/ethyl acetate = 10/1), mp 207–208 °C, 64.0 mg, 79 % yield. ^1^H NMR (500 MHz, CDCl_3_): *δ* 8.37–8.33 (m, 2H), 7.82–7.80 (m, 1H), 7.63–7.61 (m, 2H), 7.46–7.41 (m, 2H), 3.27–3.17 (m, 1H), 2.72–2.63 (m, 1H), 1.71 (s, 3H), 1.41 (t, *J* = 18.5 Hz, 3H). ^13^C{^1^H} NMR (125 MHz, CDCl_3_): *δ* 171.4, 148.7, 143.9, 141.9, 131.4, 131.3, 129.9, 127.5, 126.0, 125.8, 122.5 (t, *J* = 238.5 Hz), 121.4, 119.9, 115.6, 48.2 (t, *J* = 23.7 Hz), 45.5 (d, *J* = 3.3Hz), 30.8, 24.8 (t, *J* = 27.2 Hz). ^19^F NMR (471 MHz, CDCl_3_): *δ* −85.57–−86.28 (m, 1F), −87.26–−87.96 (m, 1F). HRMS (ESI-TOF) *m*/*z*: Calcd for C_19_H_15_BrF_2_N_2_O (M + H)^+^ 405.0408; found 405.0409. IR (KBr) ν 3420, 3074, 3004, 2946, 2713, 2359, 1910, 1793, 1725, 1543, 1420, 1292, 1072, 983, 905, 832, 716, 654 cm^−1^.

*5-(2,2-Difluoropropyl)-5-methyl-3-(trifluoromethyl)benzo [4,5]imidazo [2,1-a]isoquinolin-6(5H)-one* (**3f**). A light-yellow solid after purification by flash column chromatography (petroleum ether/ethyl acetate = 10/1), mp 151–152 °C, 63.0 mg, 80 % yield. ^1^H NMR (500 MHz, CDCl_3_): *δ* 8.63 (d, *J* = 8.0 Hz, 1H), 8.38–8.36 (m, 1H), 7.86–7.85 (m, 1H), 7.75–7.72 (m, 2H), 7.49–7.45 (m, 2H), 3.33–3.24 (m, 1H), 2.80–2.71 (m, 1H), 1.75 (s, 3H), 1.43 (t, *J* = 18.5 Hz, 3H). ^13^C{^1^H} NMR (100 MHz, CDCl_3_): *δ* 171.4, 148.1, 143.9, 140.7, 132.8 (q, *J* = 32.8 Hz), 131.5, 126.7, 126.24, 126.20, 125.6, 124.7 (q, *J* = 3.6 Hz), 124.0–123.9 (m), 122.5 (t, *J* = 240.1Hz), 120.2, 115.8, 48.1 (t, *J* = 23.5 Hz), 45.7 (d, *J* = 3.3Hz), 30.8, 24.8 (t, *J* = 27.2 Hz). ^19^F NMR (471 MHz, CDCl_3_): *δ* −62.91 (s, 3F), −85.44–−86.13 (m, 1F), −87.85–−88.51 (m, 1F). HRMS (ESI-TOF) *m*/*z*: Calcd for C_20_H_15_F_5_N_2_O (M + H)^+^ 395.1177; found 395.1179. IR (KBr) ν 3418, 3065, 2975, 2946, 2874, 2357, 1927, 1802, 1723, 1613, 1555, 1456, 1076, 991, 905, 839, 694, 546 cm^−1^.

*Methyl 5-(2,2-difluoropropyl)-5-methyl-6-oxo-5,6-dihydrobenzo [4,5]imidazo [2,1-a]isoquinoline-3-carboxylate* (**3g**). A light-yellow liquid after purification by flash column chromatography (petroleum ether/ethyl acetate = 7/1), 57.6 mg, 75 % yield. ^1^H NMR (400 MHz, CDCl_3_): *δ* 8.57 (d, *J* = 8.0 Hz, 1H), 8.38–8.36 (m, 1H), 8.18 (s, 1H), 8.13 (dd, *J* = 8.4, 1.6 Hz, 1H), 7.86–7.84 (m, 1H), 7.49–7.44 (m, 2H), 3.98 (s, 3H), 3.31–3.19 (m, 1H), 2.87–2.75 (m, 1H), 1.76 (s, 3H), 1.40 (t, *J* = 18.8 Hz, 3H). ^13^C{^1^H} NMR (125 MHz, CDCl_3_): *δ* 171.7, 166.1, 148.6, 144.1, 140.3, 132.4, 131.5, 128.7, 128.3, 126.18, 126.16, 126.14, 126.12, 122.6 (t, *J* = 238.6 Hz), 120.1, 115.8, 52.5, 48.3 (t, *J* = 23.5 Hz), 45.7 (d, *J* = 3.4 Hz), 30.8, 24.8 (t, *J* = 27.2 Hz). ^19^F NMR (471 MHz, CDCl_3_): *δ* −85.83–−86.54 (m, 1F), −87.26–−87.96 (m, 1F). HRMS (ESI-TOF) *m*/*z*: Calcd for C_21_H_18_F_2_N_2_O_3_ (M + H)^+^ 385.1358; found 385.1360. IR (thin film) ν 3417, 3080, 2980, 2848, 2354, 1922, 1843, 1717, 1614, 1550, 1475, 1359, 982, 903, 758, 565 cm^−1^.

*5-(2,2-Difluoropropyl)-5-methyl-6-oxo-5,6-dihydrobenzo [4,5]imidazo [2,1-a]isoquinoline-3-carbonitrile* (**3h**). A white solid after purification by flash column chromatography (petroleum ether/ethyl acetate = 7/1), mp 243–244 °C, 43.5 mg, 62% yield. ^1^H NMR (500 MHz, CDCl_3_): *δ* 8.59 (d, *J* = 8.0 Hz, 1H), 8.36–8.34 (m, 1H), 7.86–7.84 (m, 1H), 7.78 (s, 1H), 7.74 (d, *J* = 8.0 Hz, 1H), 7.48–7.46 (m, 2H), 3.31–3.21 (m, 1H), 2.76–2.66 (m, 1H), 1.73 (s, 3H), 1.46 (t, *J* = 19.0 Hz, 3H). ^13^C{^1^H} NMR (125 MHz, CDCl_3_): *δ* 170.9, 147.6, 143.9, 141.0, 131.4, 131.04, 131.03, 130.9, 126.7, 126.5, 126.3, 122.5 (t, *J* = 238.6 Hz), 120.3, 118.0, 115.8, 114.5, 48.1 (t, *J* = 23.3 Hz), 45.5 (d, *J* = 3.1 Hz), 30.6, 29.6, 24.8 (t, *J* = 27.1Hz). ^19^F NMR (471 MHz, CDCl_3_): *δ* −85.22–−85.93 (m, 1F), −88.52–−89.22 (m, 1F). HRMS (ESI-TOF) *m*/*z*: Calcd for C_20_H_15_F_2_N_3_O (M + H)^+^ 352.1256; found 352.1260. IR (KBr) ν 3414, 3088, 2991, 2943, 2733, 2351, 2224, 1942, 1816, 1723, 1615, 1573, 1552, 1479, 1449, 1243, 1126, 1081, 911, 846, 741, 662 cm^−1^.

*5-(2,2-Difluoropropyl)-1,5-dimethylbenzo [4,5]imidazo [2,1-a]isoquinolin-6(5H)-one* (**3i**). A white solid after purification by flash column chromatography (petroleum ether/ethyl acetate = 10/1), mp 149–150 °C, 55.1 mg, 81% yield. ^1^H NMR (400 MHz, CDCl_3_): *δ* 8.38 (d, *J* = 8.0 Hz, 1H), 8.36–7.34 (m, 1H), 7.81–7.79 (m, 1H), 7.45–7.38 (m, 2H), 7.31 (d, *J* = 8.0 Hz, 1H), 7.27 (s, 1H), 3.22 (q, *J* = 15.2 Hz, 1H), 2.72 (q, *J* = 15.2 Hz, 1H), 2.47 (s, 3H), 1.71 (s, 3H), 1.34 (t, *J* = 18.8 Hz, 3H). ^13^C{^1^H} NMR (100 MHz, CDCl_3_): *δ* 172.3, 149.8, 144.0, 141.8, 140.0, 131.4, 129.1, 127.2, 125.9, 125.8, 125.3, 122.6 (t, *J* = 240.6 Hz), 119.7, 119.6, 115.6, 48.2 (t, *J* = 23.9 Hz), 45.5, 31.1, 24.7 (t, *J* = 27.3 Hz), 21.9. ^19^F NMR (471 MHz, CDCl_3_): *δ* −85.99–−86.18 (m, 2F). HRMS (ESI-TOF) *m*/*z*: Calcd for C_20_H_18_F_2_N_2_O (M + H)^+^ 341.1460; found 341.1462. IR (KBr) ν 3410, 3065, 2991, 2963, 2854, 2720, 2361, 1909, 1843, 1796, 1716, 1619, 1559, 1242, 985, 915, 802, 659 cm^−1^.

*5-(2,2-Difluoropropyl)-1-methoxy-5-methylbenzo [4,5]imidazo [2,1-a]isoquinolin-6(5H)-one* (**3j**). A brown solid after purification by flash column chromatography (petroleum ether/ethyl acetate = 10/1), mp 165–166 °C, 49.8 mg, 70% yield. ^1^H NMR (500 MHz, CDCl_3_): *δ* 8.39–8.37 (m, 1H), 7.91–7.89 (m, 1H), 7.51 (t, *J* = 8.0 Hz, 1H), 7.43–7.39 (m, 2H), 7.12 (d, *J* = 8.0 Hz, 1H), 7.06 (d, *J* = 8.5 Hz, 1H), 4.14 (s, 3H), 3.23 (q, *J* = 15.0 Hz, 1H), 2.71 (q, *J* = 15.5 Hz, 1H), 1.72 (s, 3H), 1.32 (t, *J* = 18.5 Hz, 3H). ^13^C{^1^H} NMR (100 MHz, CDCl_3_): *δ* 172.1, 158.7, 147.7, 144.3, 142.6, 131.7, 130.3, 125.6, 125.5, 122.5 (t, *J* = 240.6 Hz), 120.5, 119.1, 115.5, 111.8, 110.4, 56.6, 48.6 (t, *J* = 23.9 Hz), 45.4, 31.5, 24.7 (t, *J* = 27.4 Hz). ^19^F NMR (471 MHz, CDCl_3_): *δ* −85.88–−86.06 (m, 2F). HRMS (ESI-TOF) *m*/*z*: Calcd for C_20_H_18_F_2_N_2_O_2_ (M + H)^+^ 357.1409; found 357.1410. IR (KBr) ν 3395, 3198, 3063, 3007, 2923, 2848, 2684, 2544, 2369, 2226, 1936, 1792, 1703, 1612, 1540, 1480, 1242, 1098, 1055, 935, 861, 803, 594 cm^−1^.

*5-(2,2-Difluoropropyl)-1-fluoro-5-methylbenzo [4,5]imidazo [2,1-a]isoquinolin-6(5H)-one* (**3k**). A yellow solid after purification by flash column chromatography (petroleum ether/ethyl acetate = 10/1), mp 153–154 °C, 55.0 mg, 80% yield. ^1^H NMR (500 MHz, CDCl_3_): *δ* 8.38–8.37 (m, 1H), 7.94–7.92 (m, 1H), 7.51 (td, *J* = 8.0, 5.0 Hz, 1H), 7.47–7.43 (m, 2H), 7.31 (d, *J* = 8.0 Hz, 1H), 7.24–7.21 (m, 1H), 3.29–3.19 (m, 1H), 2.76–2.67 (m, 1H), 1.72 (s, 3H), 1.39 (t, *J* = 19.0 Hz, 3H). ^13^C{^1^H} NMR (100 MHz, CDCl_3_): *δ* 171.6, 160.4 (d, *J* = 262.1Hz), 145.8 (d, *J* = 8.4 Hz), 144.2, 142.5, 131.8 (d, *J* = 9.6 Hz), 130.4, 126.0 (d, *J* = 14.0 Hz), 122.8, 122.5 (t, *J* = 240.6 Hz), 120.5, 115.8, 115.6, 115.5, 111.8 (d, *J* = 9.9 Hz), 48.5 (t, *J* = 23.7 Hz), 45.4, 31.3, 24.8 (t, *J* = 27.3 Hz). ^19^F NMR (471 MHz, CDCl_3_): *δ* −85.44–−86.11 (m, 1F), −87.00–−87.69 (m, 1F), −107.12–−107.16 (m, 1F). HRMS (ESI-TOF) *m*/*z*: Calcd for C_19_H_15_F_3_N_2_O (M + H)^+^ 345.1209; found 345.1211. IR (KBr) ν 3358, 3115, 3065, 2918, 2848, 2681, 2360, 1943, 1800, 1705, 1625, 1580, 1525,1452, 1345, 1240, 932, 896, 799, 765, 602 cm^−1^.

*1-Chloro-5-(2,2-difluoropropyl)-5-methylbenzo [4,5]imidazo [2,1-a]isoquinolin-6(5H)-one* (**3l**). A white solid after purification by flash column chromatography (petroleum ether/ethyl acetate = 10/1), mp 210–211 °C, 63.4 mg, 88% yield. ^1^H NMR (400 MHz, CDCl_3_): *δ* 8.39 (dd, *J* = 6.0, 3.2 Hz, 1H), 7.93 (dd, *J* = 6.0, 3.2 Hz, 1H), 7.58–7.56 (m, 1H), 7.47–7.42 (m, 4H), 3.30–3.19 (m, 1H), 2.77–2.65 (m, 1H), 1.72 (s, 3H), 1.38 (t, *J* = 18.8 Hz, 3H). ^13^C{^1^H} NMR (100 MHz, CDCl_3_): *δ* 171.4, 147.0, 143.9, 142.9, 133.5, 131.5, 130.6, 130.4, 126.3, 125.9, 125.7, 122.5 (t, *J* = 238.8 Hz), 120.7, 120.5, 115.7, 48.49 (t, *J* = 23.7 Hz), 45.76 (d, *J* = 3.5 Hz), 31.43 (s), 24.80 (t, *J* = 27.3Hz). ^19^F NMR (471 MHz, CDCl_3_): *δ* −85.21–−85.92 (m, 1F), −86.99–−87.70 (m, 1F). HRMS (ESI-TOF) *m*/*z*: Calcd for C_19_H_15_ClF_2_N_2_O (M + H)^+^ 361.0914; found 361.0916. IR (KBr) ν 3410, 3064, 3003, 2944, 2848, 2363, 2112, 1956, 1802, 1726, 1610, 1573, 1532,1456, 1239, 1093, 1045, 951, 902, 755, 575 cm^−1^.

*1-Bromo-5-(2,2-difluoropropyl)-5-methylbenzo [4,5]imidazo [2,1-a]isoquinolin-6(5H)-one* (**3m**). A white solid after purification by flash column chromatography (petroleum ether/ethyl acetate = 10/1), mp 196–197 °C, 63.2 mg, 78% yield. ^1^H NMR (500 MHz, CDCl_3_): *δ* 8.38 (dd, *J* = 6.0, 3.0 Hz, 1H), 7.93 (dd, *J* = 6.0, 3.0 Hz, 1H), 7.82 (d, *J* = 8.0 Hz, 1H), 7.49–7.44 (m, 3H), 7.33 (t, *J* = 8.0 Hz, 1H), 3.29–3.20 (m, 1H), 2.76–2.67 (m, 1H), 1.72 (s, 3H), 1.38 (t, *J* = 19.0 Hz, 3H). ^13^C{^1^H} NMR (100 MHz, CDCl_3_): *δ* 171.3, 147.1, 143.5, 143.1, 135.3, 130.8, 130.6, 126.4, 126.3, 125.9, 122.5 (t, *J* = 240.7 Hz), 121.8, 121.3, 120.8, 115.7, 48.4 (t, *J* = 23.5 Hz), 45.9 (d, *J* = 3.2 Hz), 31.5, 24.8 (t, *J* = 27.3Hz). ^19^F NMR (471 MHz, CDCl_3_): *δ* −85.15–−85.86 (m, 1F), −86.99–−87.69 (m, 1F). HRMS (ESI-TOF) *m*/*z*: Calcd for C_19_H_15_BrF_2_N_2_O (M + H)^+^ 405.0408; found 405.0412. IR (KBr) ν 3415, 3090, 3063, 3000, 2924, 2848, 2680, 2363, 1957, 1803, 1726, 1610, 1565, 1535, 1450, 1236, 1092, 1039, 951, 898, 755, 569 cm^−1^.

*5-(2,2-Difluoropropyl)-2,5-dimethylbenzo [4,5]imidazo [2,1-a]isoquinolin-6(5H)-one* (**3n**). A white solid after purification by flash column chromatography (petroleum ether/ethyl acetate = 10/1), mp 116–117 °C, 51.7 mg, 76% yield. ^1^H NMR (500 MHz, CDCl_3_): *δ* 8.37–8.35 (m, 1H), 8.32 (s, 1H), 7.83–7.81 (m, 1H), 7.45–7.40 (m, 2H), 7.37 (s, 2H), 3.22 (q, *J* = 15.5 Hz, 1H), 2.70 (q, *J* = 15.5 Hz, 1H), 2.46 (s, 3H), 1.69 (s, 3H), 1.33 (t, *J* = 18.8 Hz, 3H). ^13^C{^1^H} NMR (100 MHz, CDCl_3_): *δ* 172.4, 149.8, 143.9, 137.9, 137.2, 132.4, 131.5, 126.7, 126.1, 125.8, 125.5, 122.6 (t, *J* = 240.5 Hz), 122.0, 119.7, 115.6, 48.2 (t, *J* = 24.0 Hz), 45.3 (d, *J* = 2.2 Hz), 31.1, 24.7 (t, *J* = 27.3Hz), 20.9. ^19^F NMR (471 MHz, CDCl_3_): *δ* −86.02–−86.20 (m, 2F). HRMS (ESI-TOF) *m*/*z*: Calcd for C_20_H_18_F_2_N_2_O (M + H)^+^ 341.1460; found 341.1462. IR (KBr) ν 3417, 3115, 3073, 2923, 2850, 2671, 2369, 1943, 1716, 1613, 1547, 1495, 1335, 1238, 1172, 1012, 878, 826, 763, 555 cm^−1^.

*5-(2,2-Difluoropropyl)-1,3,5-trimethylbenzo [4,5]imidazo [2,1-a]isoquinolin-6(5H)-one* (**3o**). A white solid after purification by flash column chromatography (petroleum ether/ethyl acetate = 10/1), mp 150–151 °C, 58.0 mg, 82% yield. ^1^H NMR (500 MHz, CDCl_3_): *δ* 8.39–8.38 (m, 1H), 7.83–7.81 (m, 1H), 7.44–7.39 (m, 2H), 7.15 (s, 2H), 3.27–3.18 (m, 1H), 3.02 (s, 3H), 2.76–2.67 (m, 1H), 2.42 (s, 3H), 1.71 (s, 3H), 1.32 (t, *J* = 19.0 Hz, 3H). ^13^C{^1^H} NMR (100 MHz, CDCl_3_): *δ* 172.5, 150.0, 144.2, 141.1, 140.2, 139.7, 132.3, 130.6, 125.5, 125.4, 125.3, 122.7 (t, *J* = 240.6 Hz), 119.9, 118.4, 115.6, 48.5 (t, *J* = 24.1Hz), 45.4, 31.7, 24.7 (t, *J* = 27.3Hz), 24.6, 21.6. ^19^F NMR (471 MHz, CDCl_3_): *δ* −85.66–−85.96 (m, 2F). HRMS (ESI-TOF) *m*/*z*: Calcd for C_21_H_20_F_2_N_2_O (M + H)^+^ 355.1616; found 355.1619. IR (KBr) ν 3394, 3115, 3061, 3001, 2924, 2853, 2746, 2359, 1953, 1914, 1805, 1716, 1615, 1533, 1462, 1229, 1081, 905, 803, 615, 544 cm^−1^.

*1,3-Dichloro-5-(2,2-difluoropropyl)-5-methylbenzo [4,5]imidazo [2,1-a]isoquinolin-6(5H)-one* (**3p**). A white solid after purification by flash column chromatography (petroleum ether/ethyl acetate = 10/1), mp 216–217 °C, 58.5 mg, 74% yield. ^1^H NMR (500 MHz, CDCl_3_): *δ* 8.38–8.36 (m, 1H), 7.92–7.91 (m, 1H), 7.58 (d, *J* = 2.0 Hz, 1H), 7.48–7.44 (m, 2H), 7.40 (d, *J* = 1.5 Hz, 1H), 3.29–3.20 (m, 1H), 2.72–2.63 (m, 1H), 1.73 (s, 3H), 1.45 (t, *J* = 19.0 Hz, 3H). ^13^C{^1^H} NMR (100 MHz, CDCl_3_): *δ* 170.7, 146.3, 144.1, 143.8, 136.2, 134.4, 131.3, 130.5, 126.5, 126.1, 122.4 (t, *J* = 240.8 Hz), 120.8, 119.2, 115.6, 48.4 (t, *J* = 23.4 Hz), 45.8 (d, *J* = 3.0 Hz), 31.3, 24.9 (t, *J* = 27.1 Hz). ^19^F NMR (471 MHz, CDCl_3_): *δ* −85.15–−85.86 (m, 1F), −87.94–−88.64 (m, 1F). HRMS (ESI-TOF) *m*/*z*: Calcd for C_19_H_14_Cl_2_F_2_N_2_O (M + H)^+^ 395.0524; found 395.0527. IR (KBr) ν 3418, 3144, 3070, 3007, 2947, 2781, 2369, 1954, 1919, 1803, 1725, 1612, 1577, 1446, 1230, 1083, 1033, 983, 911, 802, 755, 618, 555 cm^−1^.

*5-(2,2-Difluoropropyl)-2,4,5-trimethylbenzo [4,5]imidazo [2,1-a]isoquinolin-6(5H)-one* (**3q**). A white solid after purification by flash column chromatography (petroleum ether/ethyl acetate = 10/1), mp 170–171 °C, 51.0 mg, 72% yield. ^1^H NMR (400 MHz, CDCl_3_): *δ* 8.35–8.33 (m, 2 H), 7.46–7.38 (m, 1H), 7.42 (pd, *J* = 7.2, 1.6 Hz, 2H), 7.18 (s, 1H), 3.34–3.11 (m, 2H), 2.62 (s, 3H), 2.41 (s, 3H), 1.80 (s, 3H), 1.38 (t, *J* = 18.8 Hz, 3H). ^13^C{^1^H} NMR (100 MHz, CDCl_3_): *δ* 173.5, 150.4, 144.1, 137.7, 137.5, 136.4, 134.6, 131.5, 125.9, 125.2, 122.9 (t, *J* = 240.3Hz), 122.9, 119.6, 115.7, 46.6, 44.9 (t, *J* = 23.5 Hz), 27.2, 24.5 (t, *J* = 27.5 Hz), 22.8, 20.6. ^19^F NMR (471 MHz, CDCl_3_): *δ* −88.43–−88.61 (m, 1F), −88.66–−88.85 (m, 1F). HRMS (ESI-TOF) *m*/*z*: Calcd for C_21_H_20_F_2_N_2_O (M + H)^+^ 355.1616; found 355.1619. IR (KBr) ν 3385, 3060, 3010, 2917, 2851, 2357, 1994, 1909, 1795, 1706, 1612, 1542, 1455, 1228, 1152, 1026, 946, 888, 799, 765, 656, 561 cm^−1^.

*3-Bromo-5-(2,2-difluoropropyl)-5,9,10-trimethylbenzo [4,5]imidazo [2,1-a]isoquinolin-6(5H)-one* (**3r**). A yellow solid after purification by flash column chromatography (petroleum ether/ethyl acetate = 10/1), mp 226–227 °C, 72.7 mg, 84% yield. ^1^H NMR (500 MHz, CDCl_3_): *δ* 8.31 (d, *J* = 9.0 Hz, 1H), 8.12 (s, 1H), 7.60–7.58 (m, 2H), 7.56 (s, 1H), 3.26–3.16 (m, 1H), 2.70–2.61 (m, 1H), 2.41 (s, 3H), 2.39 (s, 3H), 1.70 (s, 3H), 1.39 (t, *J* = 18.5 Hz, 3H). ^13^C{^1^H} NMR (125 MHz, CDCl_3_): *δ* 171.3, 148.0, 142.4, 141.7, 135.2, 135.0, 131.2, 129.9, 129.7, 127.2, 125.3, 122.4 (t, *J* = 240.7 Hz), 121.7, 120.0, 115.9, 48.2 (t, *J* = 23.8 Hz), 45.4 (d, *J* = 3.3Hz), 30.8, 24.7 (t, *J* = 27.2 Hz), 20.5, 20.4. ^19^F NMR (471 MHz, CDCl_3_): *δ* −85.57–−86.28 (m, 1F), −87.17–−87.87 (m, 1F). HRMS (ESI-TOF) *m*/*z*: Calcd for C_21_H_19_BrF_2_N_2_O (M + H)^+^ 433.0721; found 433.0723. IR (KBr) ν 3402, 3090, 2978, 2737, 2586, 2361, 2193, 1933, 1720, 1596, 1539, 1458, 1230, 1082, 948, 859, 698, 654, 562 cm^−1^.

*5-(2,2-Difluoropropyl)-5-phenylbenzo [4,5]imidazo [2,1-a]isoquinolin-6(5H)-one* (**3s**). A yellow liquid after purification by flash column chromatography (petroleum ether/ethyl acetate = 10/1), 44.2 mg, 57% yield. ^1^H NMR (400 MHz, CDCl_3_): *δ* 8.58 (dd, *J* = 8.0, 1.2 Hz, 1H), 8.26 (d, *J* = 7.6 Hz, 1H), 7.84 (d, *J* = 7.6 Hz, 1H), 7.56–7.37 (m, 4H), 7.32–7.24 (m, 3H), 7.22–7.17 (m, 3H), 3.98–3.07 (m, 1H), 3.18–3.07 (m, 1H), 1.47 (t, *J* = 18.8 Hz, 3H). ^13^C{^1^H} NMR (125 MHz, CDCl_3_): *δ* 170.3, 149.7, 144.1, 142.3, 139.8, 131.6, 131.2, 129.3, 129.1, 128.2, 128.1, 126.8, 125.93, 125.87, 125.7, 123.6, 122.8 (t, *J* = 239.8 Hz), 119.9, 115.7, 53.4, 46.0 (t, *J* = 23.5 Hz), 25.3 (t, *J* = 27.5 Hz). ^19^F NMR (471 MHz, CDCl_3_): *δ* −84.84–−85.02 (m, 1F), −85.04–−85.22 (m, 1F). HRMS (ESI-TOF) *m*/*z*: Calcd for C_24_H_18_F_2_N_2_O (M + H)^+^ 389.1460; found 389.1463. IR (thin film) ν 3427, 3064, 2921, 2847, 2720, 2350, 1943, 1829, 1717, 1612, 1552, 1445, 1366, 1108, 1043, 942, 848, 701, 624 cm^−1^.

*5-Benzyl-5-(2,2-difluoropropyl)benzo [4,5]imidazo [2,1-a]isoquinolin-6(5H)-one* (**3t**). A white solid after purification by flash column chromatography (petroleum ether/ethyl acetate = 10/1), mp 160–161 °C, 52.3 mg, 65% yield. ^1^H NMR (500 MHz, CDCl_3_): *δ* 8.36–8.34 (m, 1H), 8.29 (d, *J* = 8.0 Hz, 1H), 7.68–7.66 (m, 1H), 7.63–7.62 (m, 2H), 7.49–7.46 (m, 1H), 7.41–7.36 (m, 2H), 6.87 (t, *J* = 7.5 Hz, 1H), 6.77 (t, *J* = 7.5 Hz, 2H), 6.49 (d, *J* = 7.5 Hz, 2H), 3.53 (d, *J* = 12.5 Hz, 1H), 3.49–3.41 (m, 1H), 3.17 (d, *J* = 12.5 Hz, 1H), 2.99–2.90 (m, 1H), 1.43 (t, *J* = 19.0 Hz, 3H). ^13^C{^1^H} NMR (100 MHz, CDCl_3_): *δ* 171.2, 149.2, 143.6, 137.4, 133.2, 130.9, 130.8, 129.1, 128.0, 127.8, 127.3, 125.72, 125.69, 125.4, 124.1, 122.6 (t, *J* = 240.8 Hz), 119.6, 115.4, 51.9, 50.7, 46.5 (t, *J* = 23.8 Hz), 25.1 (t, *J* = 27.3Hz). ^19^F NMR (471 MHz, CDCl_3_): *δ* −83.32–−84.00 (m, 1F), −84.83–−85.51 (m, 1F). HRMS (ESI-TOF) *m*/*z*: C_25_H_20_F_2_N_2_O (M + H)^+^ 403.1616; found 403.1617. IR (KBr) ν 3398, 3033, 3001, 2960, 2934, 2851, 2678, 2366, 2054, 1792, 1712, 1606, 1585, 1555, 1495, 1448, 1263, 1242, 1175, 1113, 1045, 946, 833, 694, 536 cm^−1^.

*7-(2,2-Difluoropropyl)-7-methylbenzo[h]benzo [4,5]imidazo [2,1-a]isoquinolin-8(7H)-one* (**3u**). A yellow solid after purification by flash column chromatography (petroleum ether/ethyl acetate = 10/1), mp 195–196 °C, 55.6 mg, 74% yield. ^1^H NMR (500 MHz, CDCl_3_): *δ* 10.56 (d, *J* = 8.5 Hz, 1H), 8.47–8.45 (m, 1H), 8.01 (d, *J* = 8.5 Hz, 1H), 7.96–7.94 (m, 1H), 7.91 (d, *J* = 8.0 Hz, 1H), 7.84– 7.81 (m, 1H), 7.64 (t, *J* = 7.5 Hz, 1H), 7.56 (d, *J* = 8.5 Hz, 1H), 7.50–7.46 (m, 2H), 3.37–3.27 (m, 1H), 2.90–2.81 (m, 1H), 1.78 (s, 3H), 1.33 (t, *J* = 19.0 Hz, 3H). ^13^C{^1^H} NMR (100 MHz, CDCl_3_): *δ* 172.3, 149.7, 144.0, 140.6, 132.7, 132.0, 130.4, 130.3, 128.7, 128.4, 128.2, 126.9, 125.9, 125.8, 122.5 (d, *J* = 240.5 Hz), 123.7, 120.1, 117.6, 115.7, 47.9 (t, *J* = 24.1Hz), 45.9, 31.0, 24.6 (t, *J* = 27.3Hz). ^19^F NMR (471 MHz, CDCl_3_): *δ* −86.07–−86.30 (m, 1F), −86.32–−86.46 (m, 1F). HRMS (ESI-TOF) *m*/*z*: Calcd for C_23_H_18_F_2_N_2_O (M + H)^+^ 377.1460; found 377.1462. IR (KBr) ν 3405, 3104, 3058, 3003, 2940, 2726, 2356, 1940, 1789, 1715, 1619, 1572, 1526, 1453, 1305, 1266, 1182, 1078, 975, 939, 905, 823, 749, 606, 586 cm^−1^.

*4-(2,2-Difluoropropyl)-4-methylbenzo [4,5]imidazo [1,2-a]thieno [2,3-c]pyridin-5(4H)-one* (**3v**). A yellow solid after purification by flash column chromatography (petroleum ether/ethyl acetate = 10/1), mp 187–188 °C, 44.5 mg, 67% yield. ^1^H NMR (400 MHz, CDCl_3_): *δ* 8.33–8.30 (m, 1H), 7.78–7.75 (m, 1H), 7.59 (d, *J* = 4.8 Hz, 1H), 7.44–7.38 (m, 2H), 7.11 (d, *J* = 5.2Hz, 1H), 3.24–3.12 (m, 1H), 2.66–2.54 (m, 1H), 1.67 (s, 3H), 1.38 (t, *J* = 18.8 Hz, 3H). ^13^C{^1^H} NMR (125 MHz, CDCl_3_): *δ* 172.6, 146.4, 145.9, 143.9, 130.9, 130.4, 125.9, 125.8, 125.5, 123.6, 122.4 (t, *J* = 240.5 Hz), 119.7, 115.2, 48.2 (t, *J* = 24.3Hz), 45.6–45.5 (m), 29.9, 24.5 (t, *J* = 27.3Hz). ^19^F NMR (471 MHz, CDCl_3_): *δ* −86.86–−87.18 (m, 2F). HRMS (ESI-TOF) *m*/*z*: Calcd for C_17_H_14_F_2_N_2_OS (M + H)^+^333.0868; found 333.0870. IR (KBr) ν 3358, 3117, 3065, 2921, 2851, 2681, 2357, 1942, 1866, 1706, 1619, 1579, 1525, 1455, 1425, 1348, 1248, 1168, 1098, 935, 895, 831, 802, 768, 674, 599 cm^−1^.

*5-(2,2-Difluoropropyl)-3,5,12-trimethylindolo [2,1-a]isoquinolin-6(5H)-one* (**5a**). A white gummy after purification by flash column chromatography (petroleum ether/ethyl acetate = 20/1), 46.6 mg, 66% yield. ^1^H NMR (400 MHz, CDCl_3_): *δ* 8.62 (d, *J* = 7.2Hz, 1H), 7.95 (d, *J* = 8.0 Hz, 1H), 7.60–7.58 (m, 1H), 7.41–7.34 (m, 3H), 7.23 (d, *J* = 8.4 Hz, 1H), 3.28–3.16 (m, 1H), 2.71–2.60 (m, 1H), 2.65 (s, 3H), 2.44 (s, 3H), 1.69 (s, 3H), 1.31 (t, *J* = 18.8 Hz, 3H). ^13^C{^1^H} NMR (100 MHz, CDCl_3_): *δ* 172.0, 137.3, 137.0, 134.0 (d, *J* = 30.9 Hz), 132.6, 129.6 (d, *J* = 24.5 Hz), 128.3, 127.7, 125.4, 124.9, 124.2, 123.2, 122.9 (t, *J* = 240.2Hz), 118.2, 116.7, 113.6, 48.1 (t, *J* = 24.2Hz), 45.0–44.8 (m), 31.4, 24.5 (t, *J* = 27.4 Hz), 21.5, 11.5. ^19^F NMR (471 MHz, CDCl_3_): *δ* −84.11–−84.82 (m, 1F), −85.19–-85.89 (m, 1F). HRMS (ESI-TOF) *m*/*z*: Calcd for C_22_H_21_F_2_NO (M + H)^+^ 354.1664; found 354.1666. IR (thin film) ν 3358, 3184, 3071, 3001, 2923, 2850, 2567, 2363, 1919, 1716, 1627, 1570, 1489, 1392, 1320, 1158, 1091, 1012, 839, 766, 672 cm^−1^.

*5-(2,2-Difluoropropyl)-3-fluoro-5,12-dimethylindolo [2,1-a]isoquinolin-6(5H)-one* (**5b**). A yellow liquid after purification by flash column chromatography (petroleum ether/ethyl acetate = 20/1), 51.4 mg, 72 % yield. ^1^H NMR (500 MHz, CDCl_3_): *δ* 8.61 (d, *J* = 8.0 Hz, 1H), 8.03 (dd, *J* = 8.8, 6.0 Hz, 1H), 7.59 (d, *J* = 7.0 Hz, 1H), 7.42–7.35 (m, 2H), 7.17–7.11 (m, 2H), 3.28–3.18 (m, 1H), 2.63 (s, 3H), 2.62–2.55 (m, 1H), 1.69 (s, 3H), 1.38 (t, *J* = 19.0 Hz, 3H). ^13^C{^1^H} NMR (125 MHz, CDCl_3_): *δ* 171.2, 161.8 (d, *J* = 248.4 Hz), 139.8 (d, *J* = 6.9 Hz), 134.2, 132.4, 128.9, 126.9 (d, *J* = 8.3Hz), 125.7, 124.3, 122.7 (t, *J* = 240.6 Hz), 122.4 (d, *J* = 2.5 Hz), 118.3, 116.7, 114.9 (d, *J* = 21.8 Hz), 114.1, 113.9, 48.1 (t, *J* = 24.1Hz), 45.1, 31.2, 24.6 (t, *J* = 27.4 Hz), 11.4. ^19^F NMR (471 MHz, CDCl_3_): *δ* −84.17–−84.88 (m, 1F), −86.70–-87.41 (m, 1F), −112.80–−112.85 (m, 1F). HRMS (ESI-TOF) *m*/*z*: Calcd for C_21_H_18_F_3_NO (M + H)^+^ 358.1413; found 358.1415. IR (thin film) ν 3355, 3043, 3014, 2973, 2867, 2747, 2591, 2470, 2360, 1940, 1903, 1787, 1702, 1569, 1502, 1016, 975, 819, 755, 625, 589, 531 cm^−1^.

*3-Chloro-5-(2,2-difluoropropyl)-5,10,12-trimethylindolo [2,1-a]isoquinolin-6(5H)-one* (**5c**). A yellow liquid after purification by flash column chromatography (petroleum ether/ethyl acetate = 20/1), 56.8 mg, 76 % yield. ^1^H NMR (400 MHz, CDCl_3_): *δ* 8.62–8.59 (m, 1H), 7.97 (d, *J* = 8.4 Hz, 1H), 7.61–7.58 (m, 1H), 7.43 (d, *J* = 1.6 Hz, 1H), 7.41–7.35 (m, 3H), 3.28–3.16 (m, 1H), 2.63 (s, 3H), 2.67–2.55 (m, 1H), 1.69 (s, 3H), 1.38 (t, *J* = 18.8 Hz, 3H). ^13^C{^1^H} NMR (100 MHz, CDCl_3_): *δ* 171.2, 138.9, 134.2, 133.1, 132.3, 128.6, 127.6, 127.4, 126.2, 125.9, 124.5, 124.4, 122.8 (t, *J* = 240.4 Hz), 118.4, 116.7, 114.9, 48.0 (t, *J* = 23.9 Hz), 44.9, 31.2, 24.7 (t, *J* = 27.3Hz), 11.5. ^19^F NMR (471 MHz, CDCl_3_): *δ* −84.35–−85.06 (m, 1F), −86.79–-87.49 (m, 1F). HRMS (ESI-TOF) *m*/*z*: Calcd for C_21_H_18_ClF_2_NO (M + H)^+^ 374.1118; found 374.1120. IR (thin film) ν 3343, 3195, 3124, 3054, 2983, 2921, 2873, 2727, 2360, 2226, 1939, 1793, 1686, 1602, 1560, 1458, 1285, 1233, 1082, 1013, 978, 902, 811, 711, 619, 546 cm^−1^.

*3-Bromo-5-(2,2-difluoropropyl)-5,12-dimethylindolo [2,1-a]isoquinolin-6(5H)-one* (**5d**). A yellowish liquid after purification by flash column chromatography (petroleum ether/ethyl acetate = 20/1), 57.7 mg, 69% yield. ^1^H NMR (500 MHz, CDCl_3_): *δ* 8.60 (d, *J* = 7.5 Hz, 1H), 7.90 (d, *J* = 8.5 Hz, 1H), 7.60–7.58 (m, 2H), 7.52 (dd, *J* = 8.5, 2.0 Hz, 1H), 7.43–7.35 (m, 2H), 3.27–3.17 (m, 1H), 2.63 (s, 3H), 2.65–2.56 (m, 1H), 1.68 (s, 3H), 1.38 (t, *J* = 19.0 Hz, 3H). ^13^C{^1^H} NMR (125 MHz, CDCl_3_): *δ* 171.1, 139.2, 134.3, 132.3, 130.5, 130.3, 128.7, 126.4, 126.0, 124.9, 124.4, 122.7 (t, *J* = 240.5 Hz), 121.2, 118.5, 116.8, 115.1, 48.1 (t, *J* = 23.9 Hz), 44.9, 31.2, 24.7 (t, *J* = 27.3Hz), 11.5. ^19^F NMR (471 MHz, CDCl_3_): *δ* −84.40–-85.07 (m, 1F), −86.79–-87.47 (m, 1F). HRMS (ESI-TOF) *m*/*z*: Calcd for C_21_H_18_BrF_2_NO (M + H)^+^ 420.0594; found 420.0598. IR (thin film) ν 3340, 3188, 3124, 3054, 2868, 2727, 2357, 1940, 1902, 1870, 1789, 1693, 1603, 1553, 1289, 1238, 1079, 1015, 973, 903, 803, 761, 701, 618 cm^−1^.

*3-Chloro-5-(2,2-difluoropropyl)-5,10,12-trimethylindolo [2,1-a]isoquinolin-6(5H)-one* (**5e**). A yellow liquid after purification by flash column chromatography (petroleum ether/ethyl acetate = 20/1), 56.6 mg, 73% yield. ^1^H NMR (500 MHz, CDCl_3_): *δ* 8.34 (d, *J* = 8.5 Hz, 1H), 7.84 (d, *J* = 8.5 Hz, 1H), 7.30 (s, 1H), 7.25 (d, *J* = 9.0 Hz, 1H), 7.15 (s, 1H), 7.11 (d, *J* = 8.0 Hz, 1H), 3.14–3.04 (m, 1H), 2.50 (s, 3H), 2.52–2.43 (m, 1H), 2.38 (s, 3H), 1.56 (s, 3H), 1.25 (t, *J* = 19.0 Hz, 3H). ^13^C{^1^H} NMR (125 MHz, CDCl_3_): *δ* 170.9, 139.0, 134.1, 133.0, 132.5, 132.5, 128.8, 127.6, 127.4, 127.2, 126.1, 124.6, 122.7 (t, *J* = 240.4 Hz), 118.5, 116.4, 114.7, 48.1 (t, *J* = 24.0 Hz), 44.9, 31.2, 24.6 (t, *J* = 27.4 Hz), 21.6, 11.5. ^19^F NMR (471 MHz, CDCl_3_): *δ* −84.19–-84.90 (m, 1F), −86.72–−87.42 (m, 1F). HRMS (ESI-TOF) *m*/*z*: Calcd for C_22_H_20_ClF_2_NO (M + H)^+^ 388.1274; found 388.1275. IR (thin film) ν 3357, 3094, 2980, 2938, 2861, 2733, 2359, 1877, 1692, 1592, 1556, 1492, 1462, 1296, 1242, 1172, 1073, 939, 901, 801, 714, 664, 619, 591 cm^−1^.

*5-(2,2-Difluoropropyl)-12-ethyl-5-methylindolo [2,1-a]isoquinolin-6(5H)-one* (**5f**). A yellow liquid after purification by flash column chromatography (petroleum ether/ethyl acetate = 20/1), 52.9 mg, 75% yield. ^1^H NMR (500 MHz, CDCl_3_): *δ* 8.64 (d, *J* = 7.5 Hz, 1H), 8.00 (d, *J* = 8.0 Hz, 1H), 7.61 (d, *J* = 7.0 Hz, 1H), 7.47 (d, *J* = 7.5 Hz, 1H), 7.44–7.35 (m, 4 H), 3.26–3.13 (m, 3H), 2.70–2.61 (m, 1H), 1.70 (s, 3H), 1.42 (t, *J* = 7.5 Hz, 3H), 1.31 (t, *J* = 19.0 Hz, 3H). ^13^C{^1^H} NMR (125 MHz, CDCl_3_): *δ* 172.0, 137.1, 134.5, 131.8, 128.8, 127.5, 127.4, 127.3, 125.7, 124.7, 124.3, 122.9 (t, *J* = 240.2Hz), 121.1, 118.2, 116.9, 48.1 (t, *J* = 24.3Hz), 44.9, 31.3, 24.5 (t, *J* = 27.4 Hz), 18.6, 13.3. ^19^F NMR (471 MHz, CDCl_3_): *δ* −84.21–−84.92 (m, 1F), −85.41–−86.12 (m, 1F). HRMS (ESI-TOF) *m*/*z*: Calcd for C_22_H_21_F_2_NO (M + H)^+^ 354.1664; found 354.1667. IR (thin film) ν 3364, 3118, 3068, 2973, 2931, 2878, 2681, 2367, 1845, 1720, 1700, 1677, 1606, 1559, 1456, 1395, 1335, 1262, 1128, 1085, 903, 805, 736, 702, 552 cm^−1^.

*5-(2,2-Difluoropropyl)-12-ethyl-5,10-dimethylindolo [2,1-a]isoquinolin-6(5H)-one* (**5g**). A yellow solid after purification by flash column chromatography (petroleum ether/ethyl acetate = 20/1), mp 151–152 °C, 47.7 mg, 65% yield. ^1^H NMR (500 MHz, CDCl_3_): *δ* 8.50 (d, *J* = 8.5 Hz, 1H), 7.98 (d, *J* = 8.0 Hz, 1H), 7.46 (d, *J* = 7.5 Hz, 1H), 7.45–7.35 (m, 3H), 7.22 (d, *J* = 8.0 Hz, 1H), 3.25–3.16 (m, 1H), 3.13 (q, *J* =7.5 Hz, 2H), 2.69–2.60 (m, 1H), 2.51 (s, 3H), 1.69 (s, 3H), 1.41 (t, *J* = 7.5 Hz, 3H), 1.30 (t, *J* = 19.0 Hz, 3H). ^13^C{^1^H} NMR (125 MHz, CDCl_3_): *δ* 171.7, 137.0, 133.9, 132.6, 131.9, 128.9, 127.4, 127.3, 127.0, 125.8, 124.6, 122.9 (t, *J* = 240.2Hz), 121.0, 118.2, 116.6, 48.1 (t, *J* = 24.4 Hz), 44.8, 31.2, 24.4 (t, *J* = 27.4 Hz), 21.6, 18.6, 13.3. ^19^F NMR (471 MHz, CDCl_3_): *δ* −84.10–−84.77 (m, 1F), −85.39–−86.06 (m, 1F). HRMS (ESI-TOF) *m*/*z*: Calcd for C_23_H_23_F_2_NO (M + H)^+^ 368.1820; found 368.1824. IR (KBr) ν 3358, 3121, 3068, 2968, 2936, 2871, 2733, 2369, 2249, 1917, 1770, 1693, 1592, 1560, 1493, 1466, 1309, 1238, 1122, 1079, 943, 896, 816, 734, 561 cm^−1^.

*5-(2,2-Difluoropropyl)-12-ethyl-10-fluoro-5-methylindolo [2,1-a]isoquinolin-6(5H)-one* (**5h**). A yellow liquid after purification by flash column chromatography (petroleum ether/ethyl acetate = 20/1), 51.9 mg, 70% yield. ^1^H NMR (500 MHz, CDCl_3_): *δ* 8.57 (dd, *J* = 9.0, 5.0 Hz, 1H), 7.98 (d, *J* = 7.5 Hz, 1H), 7.47 (d, *J* = 7.5 Hz, 1H), 7.44–7.37 (m, 2H), 7.23 (dd, *J* = 9.0, 2.5 Hz, 1H), 7.09 (td, *J* = 9.0, 2.5 Hz, 1H), 3.24–3.15 (m, 1H), 3.12–3.07 (m, 2H), 2.69–2.60 (m, 1H), 1.69 (s, 3H), 1.40 (t, *J* = 7.5 Hz, 3H), 1.31 (t, *J* = 19.0 Hz, 3H). ^13^C{^1^H} NMR (125 MHz, CDCl_3_): *δ* 171.8, 160.4 (d, *J* = 241.1Hz), 137.3, 133.2 (d, *J* = 9.4 Hz), 130.7, 130.4, 127.65 (d, *J* = 41.6 Hz), 127.3, 125.4, 124.8, 122.8 (d, *J* = 240.4 Hz), 120.6 (d, *J* = 4.1Hz), 118.0 (d, *J* = 9.0 Hz), 113.1 (d, *J* = 24.6 Hz), 104.0 (d, *J* = 24.0 Hz), 48.3 (t, *J* = 24.2Hz), 44.8, 31.2, 24.6 (t, *J* = 27.4 Hz), 18.7, 13.2. ^19^F NMR (471 MHz, CDCl_3_): *δ* −84.78–−85.42 (m, 1F), −85.86–−86.52 (m, 1F), −118.11–−118.16 (m, 1F). HRMS (ESI-TOF) *m*/*z*: Calcd for C_22_H_20_F_3_NO (M + H)^+^ 372.1570; found 372.1575. IR (thin film) ν 3360, 3124, 3075, 3038, 2960, 2928, 2877, 2850, 2764, 2367, 2249, 2172, 2039, 1967, 1940, 1889, 1826, 1692, 1616, 1562, 1399, 1340, 1173, 976, 902, 846, 734, 691,652, 595 cm^−1^.

*3-Chloro-5-(2-cyclopropyl-2,2-difluoroethyl)-5,12-dimethylindolo [2,1-a]isoquinolin-6(5H)-one* (**5i**). A yellow solid after purification by flash column chromatography (petroleum ether/ethyl acetate = 20/1), mp 155–156 °C, 50.4 mg, 63% yield. ^1^H NMR (500 MHz, CDCl_3_): *δ* 8.61 (d, *J* = 7.5 Hz, 1H), 7.97 (d, *J* = 8.5 Hz, 1H), 7.59 (d, *J* = 8.0 Hz, 1H), 7.44 (s, 1H), 7.42–7.36 (m, 3H), 3.40–3.30 (m, 1H), 2.76–2.67 (m, 1H), 2.63 (s, 3H), 1.69 (s, 3H), 1.02–0.95 (m, 1H), 0.44–0.23 (m, 4 H). ^13^C{^1^H} NMR (125 MHz, CDCl_3_): *δ* 171.2, 139.2, 134.3, 133.1, 132.3, 128.8, 127.5, 126.1, 125.9, 124.4, 124.3, 122.4 (t, *J* = 242.3Hz), 118.4, 116.8, 114.7, 47.7 (t, *J* = 25.9 Hz), 45.0, 31.7, 16.6 (t, *J* = 28.9 Hz), 11.5, 1.3–1.2 (m). ^19^F NMR (471 MHz, CDCl_3_): *δ* −94.49–−95.10 (m, 1F), −99.95–−100.56 (m, 1F). HRMS (ESI-TOF) *m*/*z*: Calcd for C_23_H_20_ClF_2_NO (M + H)^+^ 400.1274; found 400.1279. IR (KBr) ν 3341, 3190, 3110, 3053, 3021, 2985, 2948, 2911, 2860, 2724, 2630, 2350, 2080, 1939, 1869, 1789, 1679, 1609, 1563, 1459, 1430, 1280, 1245, 1135, 1059, 1016, 925, 895, 813, 714, 639, 539 cm^−1^.

## 4. Conclusions

In summary, a novel electrochemical tandem cyclization/difluoroethylation reaction of 2-arylbenzimidazoles/2-arylindoles was reported by our group. Various CF_2_Me-substituted benzimidazo [2,1-*a*]isoquinolin-6(5*H*)-ones and indolo [2,1-*a*]isoquinolin-6(5*H*) ones could be readily synthesized in good to high yield. Additionally, it also offered a convenient protocol for the preparation of cyclopropyldifluoromethylated indolo [2,1-*a*]isoquinolin-6(5*H*) ones. Further investigation to construct other useful substituted heterocycles by electrochemical oxidative difluoroethylation is currently underway in our laboratory as well.

## Data Availability

The data underlying this study are available in the published article and its Appendix A. Deposition Numbers 2303467 (for **3m**) and 2307007 (for **5i**) contain the Appendix A for this paper. These data can be obtained free of charge via www.ccdc.cam.ac.uk/data_request/cif (accessed on 25 October 2023 and 10 November 2023), by emailing da-ta_request@ccdc.cam.ac.uk, or by contacting The Cambridge Crystallographic Data Centre, 12 Union Road, Cambridge CB2 1EZ, UK; Fax: +44 1223 336033.

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
