# Peer review of "Electrochemical Radical Tandem Difluoroethylation/Cyclization of Unsaturated Amides to Access MeCF2-Featured Indolo/Benzoimidazo [2,1-a]Isoquinolin-6(5H)-ones"

_molecules, 2024, doi:10.3390/molecules29050973_

Round 1

Reviewer 1 Report

Comments and Suggestions for Authors

Dear authors

In their manuscript, Tian Y., and co-workers report theElectrochemical Radical Tandem Difluoroethylation/Cyclization of Unsaturated Amides to Access MeCF2-Featured Indolo/Benzoimidazo[2,1-a]isoquinolin-6(5H)-ones”, it is study closely related to the published in others papers, in general, the advantages e.g., yields, process, selectivity, etc., should be discussed. (https://doi.org/10.1039/D1CC03907E, https://doi.org/10.1039/D2OB01732F, https://pubs.acs.org/doi/10.1021/acscatal.0c01324).

The overall flow and structure of the text are well written. Despite that, the following issues should be addressed in a revision.

1)    The authors should indicate the basic spectroscopic and physicochemical data of the derivatives of 1 and 4.

2)    Pag. 1, check Scheme 1-1, as well as the following

3)    Pag. 3, check, line 81, -Ar1-

4)    To see the scope of the methodology, it is important to analyze the effect when the double bond is substituted at the terminal carbon? therefore, experiments should be performed with other derivatives.

5)    The yields reported in Table 1 indicate that they are isolated, but how much was the conversion? Since they show varied yields, what is the remainder, evidence should be provided to validate the answer.

6)    The SI should describe which signals correspond to the proposed structures, indicating m (multiplicity) is not adequate when the multiplicity profile is very clear. Therefore, describe in detail each signal, analyze for example 3a.

7)    Attach HRMS chromatograms.

8)    Verify the nomenclature described, or indicate which nomenclature is used to describe your compounds.

Author Response

Question 1:The authors should indicate the basic spectroscopic and physicochemical data of the derivatives of 1 and 4.

Response: Thanks for the comments. As known compounds, the basic spectroscopic and physicochemical data of the derivatives of 1 and 4 could be found in published methods. And we have cited the corresponding reference 35.

Question 2: Pag. 1, check Scheme 1-1, as well as the following

Response: Thanks for the comments. We have corrected "Scheme" to "Figure".

Question 3:Pag. 3, check, line 81, -Ar1-.

Response: Thanks for the comments. We have corrected it in the modified manuscript.

Question 4:To see the scope of the methodology, it is important to analyze the effect when the double bond is substituted at the terminal carbon? therefore, experiments should be performed with other derivatives.

Response: Thanks for the comments. We have tried other unsaturated substrates, such as 1-(2-phenyl-1H-benzo[d]imidazol-1-yl)prop-2-yn-1-one and 3-phenyl-1-(2-phenyl-1H-benzo[d]imidazol-1-yl)prop-2-yn-1-one under the standard condition, but the results were poor and most of starting materials were detected by TLC.

Question 5:The yields reported in Table 1 indicate that they are isolated, but how much was the conversion? Since they show varied yields, what is the remainder, evidence should be provided to validate the answer.

Response: Thanks for the comments. There are some losses in the process of column chromatography treatment for the products of high yields. Some starting materials were detected by TLC for the products of lower yields.

Question 6:The SI should describe which signals correspond to the proposed structures, indicating m (multiplicity) is not adequate when the multiplicity profile is very clear. Therefore, describe in detail each signal, analyze for example 3a.

Response: Thanks for the comments. It’s hard to describe in detail each signal especially located in the aromatic region for the products. The 2D-NMR is needed to describe in detail each signal, which will be a tedious project for us. In addition, it seems to be not usual to describe in detail each signal in articles that focus on synthetic methodology.

Question 7:Attach HRMS chromatograms.

Response: Thanks for the comments. It's not the style of the journal of molecules to attach HRMS chromatograms and it is not a convention in articles that emphasize reporting synthetic methods.

Question 8:Verify the nomenclature described, or indicate which nomenclature is used to describe your compounds.

Response: Thanks for the comments. We named all the compounds by the software of ChemDraw. The nomenclature is IUPAC nomenclature.

Reviewer 2 Report

Comments and Suggestions for Authors

The presented manuscript is devoted to a development of metal-free electrochemical oxidative difluoroethylation of 2-arylbenzimidazoles for the synthesis of MeCF2-condensed heterocycles. Optimization studies and scope are fine, all target compounds were properly characterized. Some control experiments support the proposed reaction mechanism. The quality of SI is also very high. I suggest publication of this manuscript as is. My recommendation is to place conclusion section before experimental part. In addition I may suggest to include some brief discussion on X-ray diffraction study. I understand that in this case X-ray diffraction is rather routine to prove the molecular structure, but some discussion might be helpful.

Author Response

Question 1: In addition I may suggest to include some brief discussion on X-ray diffraction study. I understand that in this case X-ray diffraction is rather routine to prove the molecular structure, but some discussion might be helpful.

Response: Thanks for the comments. We have added the state of single crystal X-ray diffraction analysis in the modified manuscript.

Reviewer 3 Report

Comments and Suggestions for Authors

The reviewed paper concerns the electrochemical radical tandem difluoroethylation/cyclization of unsaturated amides. It is an interesting, well-written, well-organized paper that can be published in Molecules. However, I have some questions and suggestions listed below:

1. Consider adding a photo of your electrochemical equipment to supporting information.

2. How did you optimize the reaction time? Did you conduct the reaction also under constant current conditions? How much charge is required for the reaction?

3. Did you observe the formation of dimers like MeCF2-CF2Me or A-A (see scheme 4) during the reaction?

4. Are all the obtained compounds unknown (not described earlier in the literature)?

5. Melting points and IR data should be added.

6. NMR descriptions should be checked:

The multiplicity (sometimes, it does not correspond to the structure):

3d (45.5, d?)

3e (131.4, d?, 45.5, d?)

3j (124.8, d, J=463Hz?)

3q (7.42 pd?)

5a (134, d?... 125.2, d?)

5i (1.25, dt?) - consider 2 x t (diastereotopic groups)

Check the number of 'protons' - compound 3s

Check the number of 'carbons' - compounds 3c, 3e, 3h, 5f, 5g

Please verify the NMR frequencies, for example: 3o, 3u, and 3v (please check all the descriptions):

3o -> 1H (500 MHz) vs 13C (100 MHz) 3u-> 1H (500 MHz) vs 13C (100 MHz) 3v -> 1H (400 MHz) vs 13C (125MHz)

7. Lines 39-49 - Please change "Scheme" to "Figure"

8. Supporting Information: Please correct the titles of NMR spectra (they should be on the same page as spectra - see from page S15).

Author Response

Question 1: Consider adding a photo of your electrochemical equipment to supporting information?

Response: Thanks for the comments. We have added the photo of electrochemical equipment to the supporting information.

Question 2: How did you optimize the reaction time? Did you conduct the reaction also under constant current conditions? How much charge is required for the reaction?

Response: Thanks for the comments. We optimized the reaction time by analyzing the yields at different times. And we conduct the reaction under constant voltage conditions. The charge is hard to calculate for the reaction according to the formula (Q=It) as the current under constant voltage is constantly changing with time.

Question 3: Did you observe the formation of dimers like MeCF2-CF2Me or A-A (see scheme 4) during the reaction?

Response: Thanks for the comments. We don’t observe the formation of dimers like MeCF2-CF2Me or A-A (see scheme 4) during the reaction by the HRMS.

Question 4: Are all the obtained compounds unknown (not described earlier in the literature)?

Response: Many thanks for the comments. All the obtained compounds are unknown.

Question 5: Melting points and IR data should be added.

Response: Thanks for the comments. We have added melting points and IR data in the modified manuscript.

Question 6: NMR descriptions should be checked:

The multiplicity (sometimes, it does not correspond to the structure):

3d (45.5, d?)

3e (131.4, d?, 45.5, d?)

3j (124.8, d, J=463Hz?)

3q (7.42 pd?)

5a (134, d?... 125.2, d?)

5i (1.25, dt?) - consider 2 x t (diastereotopic groups)

Check the number of 'protons' - compound 3s

Check the number of 'carbons' - compounds 3c, 3e, 3h, 5f, 5g

Please verify the NMR frequencies, for example: 3o, 3u, and 3v (please check all the descriptions):

3o -> 1H (500 MHz) vs 13C (100 MHz)

3u-> 1H (500 MHz) vs 13C (100 MHz)

3v -> 1H (400 MHz) vs 13C (125MHz)

Response: Thanks for the comments. We checked the multiplicity for 3d (45.5, d) and observed it the double multiplicity. We think it is the influence of fluorine atom.

For 3e (131.4, d), we found it was incorrectly typed and should be corrected for 131.4, 131.3 ppm. For 3e (45.5, d), we think it is the influence of fluorine atom.

For 3j (124.8, t, J=463Hz), we found it was incorrectly typed and should be corrected for 122.5 (t, J = 240.6 Hz). We also checked the other chemical shift of this compound, some mistakes have been corrected.

For 3q (7.42 pd), we found it was incorrectly typed and should be corrected for 7.46-7.38 (m, 1H).

For 5a (134, d... 125.2, d), we found it was incorrectly typed and should be corrected for 134.2, 133.9, 125.4, 124.9 ppm.

For 5i(1.25, dt?), we found it was incorrectly typed and should be corrected for 1.3-1.2 (m).

For 3s, the number of protons at “7.32–7.24 ppm”was incorrectly typed. It is not “3” but “4”.

For 3c, the chemical shift of this compound at 118.8 ppm should be corrected for 118.89, 188.87.

For 3e, the chemical shift of this compound at 131.4 ppm should be corrected for 131.4, 131.3 ppm.

For 3h, the chemical shift of this compound at 131.0 ppm should be corrected for 131.04, 131.03 ppm.

For 5f, we carefully check the 13C NMR spectra of this compound, and the number of carbons is found to15 in experiments, which it is not agree in the theoretical values 16. We think it might be that there was an overlap of the chemical shifts of the two carbons.

For 5g, we carefully check the 13C NMR spectra of this compound, and the number of carbons is found to15 in experiments, which it is not agree in the theoretical values 16. We think it might be that there was an overlap of the chemical shifts of the two carbons.

In order to obtain a high quality NMR spectrogram, the 1H and 13C NMR spectra of 3o, 3u, and 3v, were both recorded on a Bruker advance â…¢ 400 and 500 spectrometer. We chose the better spectras in the manuscript, so 1H and 13C NMR frequencies are not correspond to each other for 3o, 3u, and 3v.

Question 7: Lines 39-49 - Please change "Scheme" to "Figure".

Response: Thanks for the comments. We have done that in the modified manuscript.

Question 8: Supporting Information: Please correct the titles of NMR spectra (they should be on the same page as spectra - see from page S15).

Response: Thanks for the comments. We have corrected it in the modified manuscript.

Round 2

Reviewer 1 Report

Comments and Suggestions for Authors

Dear Authors

I thank you for their responses, however, many of them do not show the right information.

As a recommendation: There are issues that need to be reconsidered, getting the nomenclature from Chem Draw is not necessarily valid information. Review the IUPAC documents.

As indicated, evidence should be placed on the answer sheet (question 4, 5).

I agree in part with what was indicated in the development of methodology. However, it must be described as completely as possible, from the procedure to the evidence of the product, intermediaries or impurities. TLC is not all, HPLC or CG-MS should be used.
As good laboratory practices, some points should be analyzed in detail, such as multiplicity of signals, I understand that some samples show complexity due to the proximity, but others (3a) are observed between 8.4 to 8.6 defined signals, as an example.

Best regards
